# Menstrual hygiene management interventions and their effects on schoolgirls' menstrual hygiene experiences in low and middle countries: A systematic review

**Balem Demtsu Betsu**[1]*, **Araya Abrha Medhanyie**[2], **Tesfay Gebregzabher Gebrehiwet**[2], **L. Lewis Wall**[3,4,5]

1 Department of Midwifery, College of Health Sciences, Mekelle University, Mekelle, Ethiopia, 2 School of Public Health, College of Health Sciences, Mekelle University, Mekelle, Ethiopia, 3 Department of Anthropology, College of Arts and Sciences, Washington University in St. Louis, St. Louis, MO, United States of America, 4 Department of Obstetrics & Gynecology, Washington University in St. Louis, St. Louis, MO, United States of America, 5 Department of Obstetrics & Gynecology, Ayder Comprehensive Specialized Hospital, College of Health Sciences, Mekelle University, Mekelle, Ethiopia

☯ These authors contributed equally to this work.

* balemdim@gmail.com

**Data Availability Statement:** data are all contained within the manuscript and/or Supporting Information files.

## Abstract

### Background

Inadequate menstrual hygiene management can result in physical, social, psychological, and educational challenges for schoolgirls. To address these issues, researchers have conducted intervention studies, but the impact on school attendance has varied. This review has systematically collected and evaluated evidence about the effects of menstrual hygiene interventions on schoolgirls.

### Method

A systematic search of the literature was done and reported according to the Preferred Reporting Items for Systematic Reviews and Meta-Analyses (PRISMA statement). Both peer-reviewed journals and gray literature were searched using PubMed and Google Scholar. The search included individual, or cluster randomized controlled trials, and quasi-experimental studies, and covered the period from the date of indexing until January 3, 2023.

### Result

A review of sixteen trial studies showed that menstrual hygiene interventions have a positive effect on schoolgirls' school attendance, performance, and dropout rates, as well as on their menstrual knowledge, attitudes, practices, and emotional well-being. There was a low to medium risk of bias in most of the studies. Additionally, the literature overlooked the impact of interventions that involve parental and male engagement, interventions correcting

**Funding:** The author(s) received no specific funding for this work.

**Competing interests:** LLW serves as a non-compensated member of the board of directors of the charity, Dignity Period. The other authors have no competing interests to declare. This does not alter our adherence to PLOS ONE policies on sharing data and materials.

community misperceptions about menstruation, and the impact of infrastructure improvements on water, sanitation, and hygiene.

## Conclusion

Interventions aimed at improving menstrual hygiene management can enhance schoolgirls' educational outcomes, and can improve their menstrual knowledge, attitudes, and practices by helping them manage their periods more effectively. Most interventions have focused on the provision of menstrual products and menstrual education but have neglected improvements in the physical environment at home and school and the social norms surrounding menstruation. Trial studies should take a holistic approach that considers the total socio-cultural environment in which menstrual hygiene management takes place, thus enabling stakeholders and policymakers to develop sustainable, long-term solutions to these problems.

## Introduction

Menstrual hygiene management (MHM) is defined as "women and adolescent girls using a clean menstrual management material to absorb or collect menstrual blood that can be changed in privacy as often as necessary for the duration of a menstrual period, using soap and water for washing the body as required, and having access to safe and convenient facilities to dispose of used menstrual management materials" [1]. Sommer et al. also added that menstruators should understand the menstrual cycle and be able to manage it comfortably and confidently [2].

Globally, over 50% of females are of reproductive age, and 500 million lack adequate menstrual hygiene facilities [3–5]. Many girls reach menarche without adequate knowledge or the skills to manage menstruation hygienically [6–9]. More than half (52%) of adolescent girls in Ethiopia have never received any information about menstrual hygiene [10], because of religious taboos, socio-cultural misinformation, and inadequate menstrual supplies and facilities, which leads to fear, confusion, and lack of confidence when menarche occurs [11–15].

Challenges related to menstrual hygiene have been found to contribute to absenteeism, poor academic performance, and school dropout among girls [16–19]. In Sub-Saharan Africa, a significant proportion of girls (50%-70%) miss school for 1.6–2.1 days per month due to menstruation, and more than half of girls in Ethiopia miss school during their periods [18, 20]. Menstrual hygiene challenges also have negative impacts on health, psychosocial well-being, economic opportunities, and gender equality [21–26]. These challenges include insufficient knowledge about menstruation; inadequate access to water, sanitation, and hygiene services; lack of adequate hygiene materials; and social norms unsupportive of those who menstruate [27–29].

Several programs and global initiatives, such as the 'MHM in Ten Agenda', are being implemented to enhance menstrual hygiene management among schoolgirls [30]. These programs have utilized various interventions including the provision of menstrual hygiene products and supplies, improved water sanitation and hygiene facilities, as well as increased health education.

Studies assessing the effect of these interventions on girls' school attendance, performance, physical and psychosocial wellbeing, and their knowledge and attitudes toward menstrual

hygiene management indicated varying results. Some studies have shown positive effects on school attendance, whereas others have demonstrated no effect [31–34].

One of the largest intervention studies assessing the effects of improved menstrual hygiene management on school attendance in the Tigray Region of Ethiopia, conducted during the 2015–2016 school year, demonstrated 24% fewer school absences among girls compared to boys and showed that student sex was not a predictor of school absence during a similar time-period during the previous academic year [35].

Before the search date of January 3, 2023, various systematic reviews were conducted to assess the impact of menstrual hygiene management interventions on schoolgirls. However, these reviews had different population intervention control outcome and time (PICOT) criteria, and some included studies that used cross-sectional research methods [9, 21, 22, 36]. This review specifically focuses on schoolgirls and includes up-to-date intervention studies, which distinguishes it from earlier reviews in terms of time, context, and population.

This review appraises and synthesizes the current evidence on the effects of menstrual hygiene management interventions on girls' school attendance, school performance, and school dropout, as well as their effects on emotional well-being, menstrual knowledge, attitudes, and menstrual hygiene practices.

## Materials and methods

### Study design

We conducted the review in accordance with the reporting guidelines described in the Preferred Reporting Items for Systematic Reviews and Meta-analyses (PRISMA) statement [37].

### Author's positionality

I (the first author) am a woman, a feminist, and an advocate for girls' education. I am currently pursuing a PhD in public health. I am from Ethiopia, one of the countries studied in this paper. This gives me first-hand experience of what it is like to be a menstruating schoolgirl in an LMIC. However, I am an outsider to the other countries studied in this paper, which may leave room for bias or misunderstandings in the interpretation of results. In this systematic review, the researcher's standpoint influences the research approach and findings. This study advocates accessible menstrual hygiene resources and aims to address the stigma surrounding menstruation. The conclusions are based on this perspective, and readers are encouraged to take this into account when interpreting the findings.

### Inclusion criteria

We included studies in this review based on the following criteria: participants, interventions and comparators, outcomes of interest, and type of study (study design) (PICOT) [38].

**Participants.** The search was limited to studies that measured outcomes on schoolgirls because the objective of the review was to evaluate how menstrual hygiene management intervention programs impact schoolgirls' attendance, academic performance, or dropout rates.

**Interventions and setting.** The menstrual hygiene management interventions can be categorized as interventions involving menstrual education and/or the provision of menstrual supplies.

- *Menstrual education interventions* include providing menstrual hygiene management information, puberty and reproductive health-related information, and training on the use of menstrual hygiene supplies.

- *Menstrual supply interventions* include providing menstrual hygiene materials and supplies as well as upgrading water and sanitation facilities. Menstrual hygiene materials are products used to absorb menstrual flow, such as pads, cloths, tampons, or cups. Menstrual supplies include soap and detergent, underwear, and analgesic medications for menstrual cramps. Menstrual facilities include toilets and water infrastructure, as well as private spaces for washing, changing, drying, and disposing of menstrual materials [27, 39].

**Comparator.** Schoolgirls who did not receive any of the interventions listed above.

**Outcomes.** School attendance, school performance, school dropout, emotional wellbeing, knowledge, attitudes, and menstrual hygiene practices.

**Type of study design.** Studies included individual, or cluster randomized controlled trials and quasi-experimental or non-randomized controlled trials.

**Publication type.** We included peer-reviewed journal articles and gray literature

**Exclusion criteria.** We excluded studies not available in the English language and conference abstracts.

## Source of information and search strategy

We retrieved data from PubMed and Google Scholar databases using a mix of medical subject headings (MeSH) and relevant keywords from the date of indexing to January 3, 2023 (Table 1). Citation lists and hand searches were done to locate additional citations. We limited the language to English. The search was re-run shortly before the final analysis.

## Data management and selection process

We imported the identified, eligible studies into EndNote version X5, a specific software program for managing bibliographic data. Two independent researchers reviewed the data to double-check the title and abstracts against the eligibility criteria. Studies that the two reviewers agreed upon were subjected to full-text review. Any dispute was settled by a third reviewer and consensus was sought by discussion. Full-text articles of the potentially relevant studies were then screened for the final inclusion if they met eligibility criteria.

**Table 1. PubMed search strategy.**

| |
|---|
| **Search #1:** "adolescent girls" OR "college students" OR "university student" OR "schoolgirls" OR youth OR ladies OR puberty OR feminine OR gender OR parents OR mothers OR fathers OR community |
| **Search # 2:** hygiene OR sanitizer OR sanitary OR sanitation OR washing OR soap OR "menstrual cup" OR "menstrual tampon" OR napkin OR pad OR products OR technology OR training OR "latrine access" OR toilet OR bathroom OR "menstrual hygiene" OR "Personal hygiene" OR "sanitation facilities" OR WASH OR "water supply" OR "water access" OR "water source" OR absorb OR absorbent OR "health education" OR "menstrual management" OR intervention |
| **Search #3:** "control group" OR homemade OR "worn out" OR rag OR cloth |
| **Search#4:** catamenia OR menarche OR menstruation OR menses OR "menstrual blood" OR "menstrual flow" OR "menstrual fluid" OR "menstrual period" |
| **Search#5:** absenteeism OR absent OR "academic performance" OR "school attainment" OR "school attendance" OR "school dropout" OR "school missing" OR "academic failure" OR vocation OR distract OR anxiety OR shame OR ashamed OR bullying OR mock OR embarrassment OR fear OR fearful OR distress" OR "isolation" OR "harassment" OR "intimidation" OR "confused" OR "depress OR confidence OR empower OR "menstrual health" OR "menstrual knowledge" OR "menstrual attitude" OR "menstrual practice" OR "movement restriction" OR "quality of life" OR wellbeing OR "reproductive health" OR psychology OR "mental health" OR psychosocial OR secrecy OR "self-esteem" OR shame OR empower OR understanding OR worries OR worry OR upset OR infection |
| **Search # 6:** Search #1 and Search #2 and Search #3 and Search #4 and Search #5 <br> Filters used: English language, Human |

### Data extraction process

A data extraction spreadsheet was prepared, and two reviewers extracted data manually. The spreadsheet was populated with the variables pertaining to the research question. From each study the following data were extracted; 1) Author name, 2) Year of publication, 3) Location, 4) Study design, 5) Population, 6) Sample size, 7) Duration of intervention, 8) Outcome measurement time, 9) Description of intervention, 10) Mode of intervention, and 11) Outcome of interest.

### Risk of bias in individual studies

Two authors assessed the retrieved articles for quality and potential risk of bias using the Joanna Briggs Institute (JBI) critical appraisal assessment tools for Randomized Controlled Trials and Quasi-Experimental Studies/non-randomized experimental studies [40]. The studies that scored above half of the scored value of the tools were considered for minimum risk of bias and included in the review.

### Data synthesis

We summarized data using tables, and narrative synthesis to include the type of intervention performed, characteristics of the target population, type of outcome, and a summary of the findings.

## Result

Sixteen trial studies that assessed the effect of menstrual hygiene management interventions on schoolgirls' attendance, school performance, and school dropout, as well as emotional well-being and menstrual hygiene knowledge, attitudes, and practices were reviewed. This included 15 peer-reviewed articles and one article in the gray literature (S1 Fig). The studies represented a total of 17, 910 schoolgirls, 4,612 mothers, and 4,500 fathers/guardians from Ethiopia, Kenya, Nigeria, Ghana, Uganda, Iran, Nepal, Bangladesh, and Indonesia. The studies analyzed schoolgirls aged 9 to 25 years with a sample size that ranged from 60–8,839. The studies involved menstruating, pre-menarchal, and dysmenorrhea-affected schoolgirls, as well as parents (Table 2).

The menstrual education components of the studies included puberty education, training on making reusable pads, distribution of books, magazines, posters, pamphlets, and menstrual calendars, as well as the incorporation of menstrual health topics into the school curriculum. Information was also shared through WhatsApp, face-to-face discussions, TV shows, festivals, essay competitions, storytelling, gamification, and question-and-answer sessions. These interventions were provided by trained individuals, including trained peer educators, teachers, research assistants, healthcare providers, and parents [31, 32, 34, 35, 41, 42, 44–46, 49–52]. The menstrual supply interventions included the provision of disposable and reusable pads, underwear, menstrual cups, soap, or detergent to wash menstrual pads, and the installation or improvement of water, sanitation, and hygiene (WASH) facilities (Table 2).

The reported trials evaluated several outcomes of interest, including school attendance, school dropout, and academic performance, as well as menstrual hygiene knowledge, attitudes, and practices, physical health, and emotional health including menstruation-related fear, shame, and stigma, as well as social attitudes regarding expected gender-related behavior (gender norms). Montgomery et al. suggested using school attendance and dropout rates as a proxy indicator of academic performance [32]. Some trials relied on self-reported/recorded attendance, which may have introduced recall bias, while others cross-checked attendance

**Table 2. Characteristics, population, intervention, and outcome of studies included in the review.**

| Author name, year, and location | Study design | Population (P) and Sample size (SS) | Duration of intervention (DOI) and Outcome measurement time (OMT) | Intervention Description (DI) and mode of intervention (MOI) | Outcome of interest |
|---|---|---|---|---|---|
| Abedian et al. 2011 Mashhad, Iran [41] | Randomized controlled trial | **P:** 19–25-year-old Dysmenorrheic University girls **SS:** Planed SS: 209 Actual SS: 165 (Peer-led education group n = 54; Health provider-led education group n = 50; Control group n = 61) | **DOI:** At baseline and two consecutive menstrual cycles (approximated to two months) **OMT:** Immediately after intervention | **ID:** self-care education Arm 1: received health provider-led self-care education Arm 2: received peer-led self-care education **MOI:** Small group discussions about self-care education held by health providers and peer educators | • The mean score of menstrual knowledge significantly increased in both groups compared to the control group (the peer-led self-care group increased by 2.1 times and health-provider 2.5 times) • Negative concepts of mean menstrual attitude decreased in the peer-led self-care education group (56.6 vs. 40.2, p = 0.009) more than the health-provider-led self-care education group (56.9 vs. 48.3, p = 0.035). • The severity of dysmenorrhea decreased between the intervention arms and control arm but not significantly between the intervention groups |
| Agbede et al. 2021 Ogun State, Nigeria [42] | Quasi-experimental | **P:** 10–19-year-old rural school adolescent girls **SS:** 120 (30 in each of 4 study arms) | **DOI:** 4 weeks (number and length of sessions not indicated) **OMT:** Immediately post-intervention (at 4 weeks) and 6 weeks follow-up | **ID:** Health education related to menstrual hygiene practice • Arm 1: peer-led education intervention • Arm 2: parent-led intervention • Arm 3: a combination of both • Arm 4: Placebo | Menstrual hygiene practices of the three intervention arms have significantly improved both in the 4th (immediate post-intervention) and in the 6th week follow-up. • While the third arm (combination of peer and parent recorded the highest mean score of practice |
| Austrian et al. 2019 Kenya [43] | Cluster-randomized controlled trial (With Four arms) | **P:** 10–21-year-old girls **SS:** 3,276 schoolgirls | **DOI:** 25 sessions each lasting for 65–95 minutes for 18 months (Weekly in 2017 and every two weeks in 2018) **OMT:** after 18 months (immediately after completion of the intervention) | **ID:** Arm 1: No intervention Arm 2: Disposable sanitary pad intervention Arm 3: Reproductive health education Arm 4: sanitary pad and reproductive health education The sanitary pad and education include: one pack of Nia Teen disposable sanitary pads distributed monthly with pairs of underwear provided once per term The reproductive health education includes puberty, gender, gender, power, and rights, being true to yourself **MOI:** Trained facilitators provided facilitated health education (FHE) and distribution of health magazine developed by ZanaAfrica based on the UNESCO International Technical Guidance on Sexuality Education incorporating gender and power in sexuality and HIV education | • Provision of Pads improved menstrual hygiene management • RH education led to improved SRH knowledge, self-efficacy, gender norms, and attitudes toward menstruation • The combined intervention had stronger impacts on reducing shame/stigma around menstruation • None of the interventions had an impact on education outcomes like school attendance and enrolment for the subsequent grade |

*(Continued)*

**Table 2.** (*Continued*)

| Author name, year, and location | Study design | Population (P) and Sample size (SS) | Duration of intervention (DOI) and Outcome measurement time (OMT) | Intervention Description (DI) and mode of intervention (MOI) | Outcome of interest |
|---|---|---|---|---|---|
| Babapour et al. 2022 Sari, northern Iran [44] | Quasi-experimental non-randomized controlled trial | **P:** 11th-grade single students with regular menstruation **SS: 90** (30 in each of the three arms) | **DOI:** Six, one-hour sessions twice a week in WhatsApp messenger. **OMT:** Not indicated | **ID**: The education sessions included: menstruation and menstrual disorders including PMS and measures to alleviate, life skills, female reproductive system • Arm 1: received education from peers • Arm 2: received education from a healthcare provider • Arm 3: is the control group **MOI** • Education is held using WhatsApp messenger • All three groups received routine school counseling. • Education providers individually uploaded a pre-prepared audio files with the related PowerPoint file in each session and allowed participants to ask questions. At the end of each session, the healthcare provider/peer asked questions about the topics and motivated to participate in the discussion. | **Primary outcome:** Premenstrual syndrome (PMS) • PMS score decreased in the intervention groups compared to the control group. • The effect size in the education by a health care provider group (Partial Eta Squared = 0.82, p < 0.0001) was more than the education by peers' group (Partial Eta Squared = 0.67, p < 0.0001). **Secondary outcomes**: General health and premenstrual dysphoric disorder • The mean score of general health (a measure of emotional distress) significantly decreased in the education group by peers (Cohen's d = 0.25, p<0.0001) and education by health care provider group Cohen's d = 0.37, p<0.0001) compared with the control group. • The intervention did not significantly reduce the frequency of premenstrual dysphoric disorder among the two intervention groups as compared to the control group (p>0.050). |
| Belay et al. 2020 Tigray Ethiopia [35] | Quasi-experimental | **P:** Grade 7–12 students **SS:** 8,839 Students in 15 intervention schools | **DOI**: one academic year **MOT**: immediately post-intervention | **ID:** Menstrual education provided to boys and girls • Girls were provided with menstrual hygiene kits containing four locally produced, reusable menstrual pads and two pairs of underwear. **MOI**: School-based distribution of a booklet called Growth and Changes, written in English and Tigrinya (the local language). • Students are encouraged to take the booklet home with them to share with their families. • Additional oral instruction was provided on-site by project staff from Mekelle University • Interactive question-and-answer sessions • Distribution of 12 211 pamphlets • Distribution of menstrual kit • Demonstration of how to use sanitary pads for girls | Girls had 24% fewer absences as compared to the control arm during the post-intervention period. |

(*Continued*)

**Table 2.** (Continued)

| Author name, year, and location | Study design | Population (P)and Sample size (SS) | Duration of intervention (DOI) and Outcome measurement time (OMT) | Intervention Description (DI) and mode of intervention (MOI) | Outcome of interest |
|---|---|---|---|---|---|
| Blake et al. 2017; Oromia Ethiopia [45] | Cluster-randomized study triangulated with a qualitative approach | **P:** Grade 6 &7 schoolgirls **SS:** 636 | **DOI:** Puberty book provided to the girls for 4 weeks **OMT:** Four weeks after the distribution of the book (no follow-up in between) | **ID:** The Ethiopian version of the girl's puberty book Growth and Changes. The book targeted girls aged 10 to 14 years, covering puberty education, menstruation and menstrual hygiene management; and culturally tailored stories. **MOI:** Book delivered to the study participants to read them. | The intervention had a positive effect on: • The girls' knowledge about menstruation with effect size of 0.6 (medium effect size • Post-intervention, girls in the intervention group were less likely to indicate that they felt fear regarding menstruation (OR = 0.70, 95% CI = [0.51, 0.95]) or shame (OR = 0.61, 95% CI = [0.38, 0.96]) than girls in the control group. |
| Fakhri et al. 2013 Mazandaran province, Iran [31] | Quasi-experimental (Non-randomized controlled cluster trial | **P:** 14 -18-year-old-girls with low socio-economic status from urban and rural public high schools **SS:** 689 (349 intervention group and 349 control group) | **DOI:** (20 hrs.) 10 sessions of 2 hr. each (Not indicated for how long) **OMT:** At the end of the education intervention | • **ID:** Training about: • personal health and hygiene during Menstruation• Significance of adolescence, physical and emotional changes during adolescence,• Pubertal and menstruation health and premenstrual syndrome**MOD:** Intervention provided by the Youth and School Health Department to the intervention arm | • **Menstrual health** especially bathing and genital hygiene improved (61.6% in the experimental group compared with 49.3% in the control group engaged in usual bathing during menstruation (p = 0.002)) • **Attitude towards menstruation** was also significantly related to menstrual health. |
| Nyadoy et al. 2022 Uganda [46] | Randomized Controlled Trial | **P:** primary school adolescent girls who reached menarche **SS:** 60 (30 control and 30 intervention group | **DOI:** One-hour session twice a week, after classes for six weeks **OMT:** Outcome assessed immediately after the intervention ended | **ID:** Menstrual health management storying and gamification **MOI:** Storying involved Senior Women Teachers and other invited role models sharing stories about the facts and myths of menstruation and menstrual hygiene management. The games involved competitive ball games such as soccer, netball, and rope work | • Girls in the treatment group (t = 8.498, df = 29, p < .05) obtained significantly higher scores (in four courses, English language, Mathematics, Integrated Science, and Social Studies) than those in the control group • The experiment group reported positive attitudes and expressed feelings of liberation from fear of boys during menstruation, |
| Oster et al. 2011 Chitwan District, Nepal [34] | Randomized controlled trial | **P:** Grade 7 and 8 schoolgirls (25 girls assigned to treatment group from each school) **SS:** 198 | **DOI:** One school-year intervention **OMT:** Outcome assessed immediately after the intervention | **ID:** Menstrual cup branded as Moon-cup **MOI:** Treatment girls and their mothers were provided with menstrual cups and instructions on how to use them. Girls were provided with a booklet of time diaries that included a menstrual calendar on which they were to note the start and end date of their period in each month. | The menstrual cup does not significantly increase school attendance |

(*Continued*)

**Table 2.** (Continued)

| Author name, year, and location | Study design | Population (P) and Sample size (SS) | Duration of intervention (DOI) and Outcome measurement time (OMT) | Intervention Description (DI) and mode of intervention (MOI) | Outcome of interest |
|---|---|---|---|---|---|
| Paul Montgomery 2012 Ghana [32] | Non-randomized-controlled trial | **P:** 12–18-year-old schoolgirls **SS:** 120 | **DOI:** Five months **OMT:** At the third and fifth month (at the end of the intervention) | **ID:** Provision of one pair of underwear and twelve pads per month for the duration of the study with instruction and demonstrations on how to use and dispose of the sanitary pads. Puberty educational about the development of secondary sex characteristics, menstruation, pregnancy, hygiene, and menses management Arm-1: Pads + puberty education Arm-2: Puberty education Arm-3: Control **MOI:** ➤ Trained research assistants provided puberty education ➤ All participants received a daily calendar, pencil, and sharpener to record their menstrual cycles | **Arm-1** (pad + education): school attendance improved significantly among participants, (lambda 0.824, F = 3.760, p, .001) **Arm-2:** education only resulted in a similar school attendance level (M = 91.26, SD = 7.82) all of which were higher than control (M = 84.48, SD = 12.39). The effect size, partial eta-squared, was 0.094. |
| Paul Montgomery et al. 2016 Uganda [47] | Cluster Quasi-Randomized Controlled Trial | **P:** Grade 3–5 schoolgirls **SS:** 356 pre and post-menarcheal girls) from 8 rural schools | **DOI:** Single session of puberty education and two times of pad distribution and soap (one sachet, 45gram (18 months apart) The education session lasted for 1.25hrs **OMT:** two years later | **ID:** provision of reusable pad and Puberty education about menstruation, early pregnancy, life skills, prevention of HIV, strategies for avoiding sexual assault, healthy relationships, and friendship formation and goal setting. **Arm-1:** puberty education **Arm-2:** provided with reusable pad 3 pairs of underwear, one sachet, and 45 grams of soap with which to wash the pads. **Arm-3:** puberty education and reusable sanitary pad **Arm-4:** A control condition | • Control schools had 17.1% (95%CI: 8.7–25.5) greater drop in school attendance than those in any intervention school • No psychosocial change was observed among the study arms |
| Phillips-Howard et al. 2016 Gem District Kenya [48] | Cluster randomized controlled feasibility study open-level RCT | **P:** 14–16 years old girls (with no precluding disability) who experience at least three menses **SS:** **Planed SS:** 3165 **Executed SS:** (644 analyzed) from 30 rural primary schools | **DOI:** 15 months **OMT:** at the end of the follow-up (intervention) | **ID:** Girls in all arms received puberty and hygiene training; hand-washing soap; and pencils for calendar completion. **Arm 1:** received one menstrual cup with written and verbal instructions on how to insert and clean **Arm 2:** received 16 disposable pads and relevant instructions. **Arm 3:** Control **MOI:** Nurses provided menstrual product-specific training from study nurses after enrolment | •**School dropout (primary outcome):** Cups or pads did not reduce school dropout (control = 8.0%, cups = 11.2%, pads = 10.2%) •**Absence:** This could not be analyzed because self-reported school absences were very rarely reported. •**STI (secondary outcome):** Lowered prevalence of C. trachomatis and T. vaginalis but not N. gonorrhea. The greatest impact was among girls who had been exposed to intervention for at least 9 months or 12 months. • Prevalence of all STIs at the end-line survey was 7.7% in the control arm versus 4.3% in the pooled cups +pads arms •**RTI:** Bacterial vaginosis was lower in the cup arm (not significant), but not in the pad arm. •**TSS:** No case reported |

*(Continued)*

**Table 2.** (Continued)

| Author name, year, and location | Study design | Population (P)and Sample size (SS) | Duration of intervention (DOI) and Outcome measurement time (OMT) | Intervention Description (DI) and mode of intervention (MOI) | Outcome of interest |
|---|---|---|---|---|---|
| Rezaei, et al. 2022 Iran [52] | Quasi-experimental study | **P:** 13–16- year-old high school students and their mothers **SS: Control**: 111 (56 students and 55 mothers) **Intervention**: 112 (58 students and 57 mothers) | **DOI:** Not indicated **OMT:** Immediately after intervention and three months later | **ID:** Educational intervention based on the PRECEDE model provided. Adolescence, puberty, menstrual cycle, abnormal signs, and common problems associated with menstruation, menstrual health, exercise, nutrition, mobility, and pain control in menstruation **MOI:** The education was provided in 3 sessions of two hours each using lecture, face-to-face discussion, and question/answer methods for students and mothers in the intervention arm | • The mean score of menstrual health behavior was significantly higher in the intervention group than in the control group, immediately (P < 0.001), and three months after intervention (P = 0.02) • Mothers' knowledge, attitude, and practice regarding menstrual health behaviors were significant reinforcing factors among the intervention group compared to the control group |
| Setyowati et al.2019 Indonesia [49] | quasi-experimental pre and post-test with a control group design | **P:** 9-12-year-old schoolgirls who had not yet experienced menarche **SS:** 174 girls | **DOI**: Not indicated **OMT:** Not indicated | **ID:** Booklet containing information about preparation for menarche, reproductive organs, physical changes during adolescence, problems during menstruation and how to deal with it, and menstrual hygiene **MOI**: Distribution of booklet to the intervention group | • Increased menstrual knowledge (OR = 45.1; 95% CI: 13.8–148.1) • Positive emotional response (OR = 12.7; 95% CI: 5.6–28.5) • Positive attitude towards menstruation (OR = 12.4; 95% CI: 5.8–26.6) |
| Sol et al. 2017 Bangladesh [50] | Cluster randomized impact evaluation | **P:** Junior secondary school girls **SS:** planned SS: 3862 girls Actual SS: 2127 (595 treatment-1, 570, treatment-2 and 962 control group) 4,500 mothers/guardians and 4,500 fathers/guardians attended the Household education sessions | **DOI:** At least twice a month from 2017–2019 **OMT:** Two years later (after the intervention) | **ID:** Construction and maintenance of menstrual health-friendly toilet facilities at school. Incorporating puberty- and menstrual health modules in the school curriculum,A 2-day session to increase menstrual health knowledge and understanding of the benefits of safe menstrual hygiene was produced for parents /guardians**Arm 1:** schools receiving a school program**Arm 2:** schools receiving a school program combined with a targeted household program ('combined program')**Arm 3:** control schools**MOI:** An extensive campaign to familiarize teachers, students, and parents, next to festivities, Group discussions, essay writing competitions, and screening of a TV-shows and extracurricular activities | **Primary outcome:** educational outcomes, psychosocial outcomes, and empowerment of adolescent girls. • Absence rates in treatment schools are significantly lower than in the control schools (no significant difference between the school program and combined program schools) • School dropout was reduced in both treatment groups as compared to controls **Secondary outcomes**: • Increase in the knowledge of girls about menstruation and menstrual health (on both treatment arms • Lowered restrictive beliefs surrounding the mobility of girls on their menses. (On both treatment arms) • More likely to get permission to go to the toilet when they ask their teacher • No treatment effects on teasing during menstruation |

(*Continued*)

**Table 2.** (Continued)

| Author name, year, and location | Study design | Population (P) and Sample size (SS) | Duration of intervention (DOI) and Outcome measurement time (OMT) | Intervention Description (DI) and mode of intervention (MOI) | Outcome of interest |
|---|---|---|---|---|---|
| Wilson et al. 2014 Rural Kenya [51] | Cluster randomized control | **P:** Schoolgirls **SS:** 302 (143 intervention and 159 control) | **DOI:** One session **OMT:** One month after intervention | **ID:** Training on how to make a reusable sanitary pad and provision of equipment to make three reusable pads. • rovide printed hand-out, as a reminder on how to make the pad and instructions about washing and drying, risk of infection or irritation of damp or poorly washed pad; with suggested ways to dry the pad outside and avoid embarrassment. • Did not include general menstrual health education to evaluate the mere effect of pad use **MOI:** training and provision of handout | • The mean number of days of school missed decreased or stayed constant among the treatment group while schools in the control group either stayed constant or increased |

records [32, 35, 48, 51] (Tables 3 and 4). The trial by Sol et al. cross-checked attendance of the school record with survey data using spot checks [50]. Official school attendance records were supplemented by individual diaries filled out by the schoolgirls in one of the trials [34]. Moreover, despite the presence of a standardized menstrual attitude questionnaire that can be validated contextually [53], some trials used non-standardized evaluation tools developed by specific researchers [31, 42, 45, 52]. Causation is difficult to determine in some studies because the sample size was quite small [as few as 60 participants [46]].

Montgomery et al. carried out a study on peri-urban schools that were comparable, but they also incorporated a remote rural site lacking experience in using sanitary pads, with no access to electricity and unpaved roads. This may have affected baseline similarity and intervention fidelity, making it harder to determine the intervention's actual effects [32]. In some studies, only a single-session educational intervention was provided, and only half of the girls attended the educational session in the study by Montgomery et al. [47, 51]. Some studies did not provide information on the follow-up or dropout rates among the study groups [31, 35, 41, 43, 49], while others achieved a statistically significant improvement in school attendance and school performance using small sample sizes [32, 46]. Blinding with respect to outcomes is crucial in reducing bias in experimental studies, but it is sometimes impractical to blind study participants, intervention providers, and assessors. This may lead to exaggerated intervention-effect estimates and performance bias [54]. Despite this, there are ongoing controversies concerning the advantage of blinding in clinical trials [55–57]. Outcome assessors, laboratory staff, and statisticians were blinded in only two trials [48, 50].

## The effect of menstrual hygiene management interventions on

**Schoolgirls' school attendance, performance, and dropout.** Of the trials that evaluated the effect of menstrual hygiene management interventions on school attendance, performance, and dropout, six studies reported that the intervention had a positive effect [32, 35, 46, 47, 50, 51] while the remaining three studies found no significant effect [34, 43, 48].

Implementing pad and education interventions together increased school attendance by almost 6 days per term, equivalent to 9% of a girls' school year. The study also showed that

**Table 3. Joanna Briggs Institute (JBI) risk of bias assessment for quasi-experimental studies.**

| JBI Critical Appraisal Checklist for Quasi-Experimental Studies | Agbede CO 2021 | Babapour et al. 2022 | Belay et al. 2020 | Chiou et al 2007 | Darabi 2022 | Fakhri 2012 | Paul Montgomery 2012 | Paul Montgomery et al 2016 | Rezaei 2022 | Scott 2009 | Setyowati et al. 2019 | Yilmaz 2019 |
|---|---|---|---|---|---|---|---|---|---|---|---|---|
| 1. Is it clear in the study what is the "cause" and what is the "effect" (i.e., there is no confusion about which variable comes first)? | Yes | Yes | Yes | No | Yes | Yes | No | Yes | Yes | No | Yes | Yes |
| 2. Were the participants included in any comparisons similar? | Yes | Un | Yes | Yes | Un | Yes | Yes | Yes | Yes | Un | Yes | No |
| 3. Were the participants included in any comparisons receiving similar treatment/care, other than the exposure or intervention of interest? | Un | Yes | Un | No | Un | Yes | Yes | Yes | Yes | Yes | Yes | Un |
| 4. Was there a control group? | Yes | Yes | Yes | Yes | Yes | Yes | Yes | Yes | Yes | Yes | Yes | Yes |
| 5. Were there multiple measurements of the outcome both pre and post-intervention/exposure? | Yes | Yes | Yes | Yes | Yes | Yes | Yes | Yes | Yes | Yes | Yes | No |
| 6. Was follow-up complete and if not, were differences between groups in terms of their follow-up adequately described and analyzed? | Yes | Yes | Un | Un | Un | Un | No | Yes | Un | Un | Un | Un |
| 7. Were the outcomes of participants included in any comparisons measured in the same way? | Yes | Yes | Yes | Yes | Un | Yes | Yes | Yes | Yes | Yes | Yes | Un |
| 8. Were outcomes measured in a reliable way? | Yes | Yes | Yes | Yes | Yes | Yes | Yes | Yes | Yes | Un | Yes | Yes |
| 9. Was appropriate statistical analysis used? | Un | Yes | Yes | Un | Un | Yes | Un | Yes | Yes | Un | Yes | Yes |
| **Overall appraisal:** | 7 | 8 | 7 | 4 | 4 | 8 | 6 | 9 | 8 | 4 | 8 | 4 |

puberty education alone improved school attendance levels five months after intervention [32].

Belay et al. carried out an intervention at 15 schools (5 rural and 10 urban) involving 8,839 students, both male and female. All students received an educational intervention concerning menstruation and the female students also received four reusable pads and two pairs of regular underwear. School attendance was analyzed before and after the intervention and compared with attendance data from the prior school year. After the intervention, girls had 24% fewer school absences than boys and student sex was not a predictor of school absence during a similar time period during the previous academic year [35]. In another study, providing reusable pads, soap, and puberty education; training on making reusable sanitary pads and providing the equipment necessary to make pads resulted in decreased school absenteeism compared to controls [47].

**Table 4. Joanna Briggs Institute (JBI) risk of bias assessment for Randomized Controlled Trials (RCT).**

| JBI Critical Appraisal Checklist for RCT | Abedian et al. 2011 | Alexander et al. 2018 | Austrian et al. 2019 | Blake et al. 2017 | Djalalinia 2012 | Kokiwar 2020 | Mbizvo 1997 | Mohammadzadeh et al 2002 | Nyadoy et al. 2022 | Oster 2011 | Phillips-Howard 2016 | Sol et al. 2017 | Wilson, et al. 2014 |
|---|---|---|---|---|---|---|---|---|---|---|---|---|---|
| 1. Was true randomization used for the assignment of participants to treatment groups? | Yes | Yes | Yes | Yes | Yes | Yes | No | Yes | Yes | Yes | Yes | Yes | Yes |
| 2. Was allocation to treatment groups concealed? | Un | Un | Un | Un | Un | Un | Un | Un | Un | Un | No | Un | No |
| 3. Were treatment groups similar at the baseline? | Yes | No | Yes | Un | Yes | Un | No | Yes | Yes | Yes | Yes | Yes | Yes |
| 4. Were participants blind to treatment assignment? | Un | Un | Un | No | Un | Un | Un | Un | Un | Un | No | Un | No |
| 5. Were those delivering treatment blind to treatment assignment? | Un | No | No | No | Un | No | Un | Un | Un | No | No | Un | Un |
| 6. Were outcomes assessors blind to treatment assignment? | Un | Un | No | Un | Un | Un | Un | No | Un | No | Yes | Un | No |
| 7. Were treatment groups treated identically other than the intervention of interest? | Yes | Yes | Yes | Yes | Un | Un | Yes | Un | Yes | Yes | Yes | Yes | Yes |
| 8. Was follow-up complete and if not, were differences between groups in terms of their follow-up adequately described and analyzed? | Un | Un | Un | Yes | Un | No | Yes | Yes | Yes | Yes | Yes | Yes | Yes |
| 9. Were participants analyzed in the groups to which they were randomized? | Yes | Yes | Yes | Yes | Un | No | Yes | Yes | Yes | Yes | Yes | Yes | Yes |

(*Continued*)

**Table 4.** (Continued)

| JBI Critical Appraisal Checklist for RCT | Abedian et al. 2011 | Alexander et al. 2018 | Austrian et al. 2019 | Blake et al. 2017 | Djalalinia 2012 | Kokiwar 2020 | Mbizvo 1997 | Mohammadzadeh et al 2002 | Nyadoy et al. 2022 | Oster 2011 | Phillips-Howard 2016 | Sol et al. 2017 | Wilson, et al. 2014 |
|---|---|---|---|---|---|---|---|---|---|---|---|---|---|
| 10. Were outcomes measured in the same way for treatment groups? | Yes | Un | Yes | Yes | Un | Yes | Yes | Yes | Yes | Yes | Yes | Yes | Yes |
| 11. Were outcomes measured in a reliable way? | Yes | Yes | Yes | Un | Un | No | No | Yes | Yes | Yes | Yes | Yes | Yes |
| 12. Was appropriate statistical analysis used? | Yes | Yes | Yes | Yes | Un | Yes | No | Un | Yes | Un | Yes | Yes | Yes |
| 13. Was the trial design appropriate, and any deviations from the standard RCT design (individual randomization, parallel groups) accounted for in the conduct and analysis of the trial? | Yes | Yes | Yes | Yes | Un | Yes | Yes | Un | Yes | Yes | Yes | Yes | Yes |
| **Total score** | 8 | 6 | 8 | 7 | 2 | 4 | 5 | 6 | 9 | 8 | 10 | 9 | 9 |

School and household-based menstrual hygiene management interventions implemented in 148 schools involving 2,127 schoolgirls showed reduced school absences and dropout rates. The interventions included the integration of menstrual health education into the school curriculum and building/maintaining water, sanitation, and hygiene (WASH) infrastructure at schools. The provision of household interventions to improve parents' knowledge didn't show significant differences between the household-based and school-based intervention arms [50] (Table 2).

On the other hand, three trials showed that school-based interventions did not affect school attendance or dropout. According to Phillips-Howard et al. providing either menstrual pads or cups did not reduce school dropout (control = 8.0%, cups = 11.2%, pads = 10.2%), though the finding should be taken with caution because of a nearly 40% loss-to-follow-up. Likewise, in this study school absenteeism was not analyzed because it was rarely reported [48]. The trial by Oster et al. found that providing menstrual cups to girls did not significantly increase school attendance [34]. This is congruent with the study by Austrian et al. that found no significant effect on school attendance after the provision of disposable sanitary pads and reproductive health education [43] (Table 2).

**Menstrual knowledge, attitude, menstrual hygiene practice, and emotional wellbeing.** Studies have shown that both menstrual education and menstrual hygiene supply interventions have a positive effect on menstrual hygiene-related knowledge, attitudes, and practices among schoolgirls (Table 5). These interventions help reduce menstrual-related shame,

**Table 5. Summary of outcome results.**

| Author | Type of intervention | Measured outcomes | | | | | | | |
|---|---|---|---|---|---|---|---|---|---|
| | | School attendance | School performance | School dropout | Menstrual knowledge | Menstrual attitude | Menstrual Practice | Emotional wellbeing | Physical health |
| Abedian et al. 2011 Mashhad, Iran [41] | Self-care education | | | | + | + | | | + |
| Agbede et al 2021 Ogun State, Nigeria [42] | Health education related to menstrual hygiene practice | | | | | | + | | |
| Austrian et al. 2019, Kenya [43] | Disposable sanitary pad; reproductive health education; sanitary pad plus reproductive health education | 0 | | | + | + | | | |
| Babapour et al. 2022 Sari, Northern Iran [44] | Education delivered by peers and by healthcare provider | | | | | | | + | 0 |
| Belay et al. 2020 Tigray, Ethiopia [35] | Provision of menstrual education | + | | | | | | | |
| Blake et al. 2017 Oromia, Ethiopia [45] | Delivery of puberty book (Growth and Changes) | | | | + | + | | | |
| Fakhri et al. 2013 Mazandaran province, Iran [31] | Providing puberty and menstrual education | | | | | + | + | | |
| Nyadoy et al. 2022 Uganda [46] | Menstrual health management story letting and games | | + | | | | | | |
| Oster et al. 2011 Chitwan District, Nepal [34] | Delivering menstrual cup | 0 | | | | | | | |
| Paul Montgomery 2012 Ghana [32] | Delivering disposable pads and puberty education | + | | | | | | | |
| Paul Montgomery et al. 2016 Uganda [47] | Provision of reusable pads and puberty education | + | | | | | | 0 | |
| Phillips-Howard et al. 2016 Gem District, Kenya [48] | Delivering puberty and hygiene training; hand-washing soap; and pencils for calendar completion | | | 0 | | | | | + |
| Rezaei, et al. 2022 Iran [52] | Provision of adolescence, puberty, and menstrual education | | | | + | + | + | | |
| Setyowati et al.2019 Indonesia [49] | Provision of a booklet about preparation for menarche, reproductive organs, and physical changes during adolescence | | | | + | + | | + | |
| Sol et al. 2017 Bangladesh [50] | Construction and maintenance of menstrual health-friendly toilet facilities at school. Incorporating puberty- and menstrual hygiene modules into the school curriculum | + | | + | + | + | | | |
| Wilson et al. 2014 Rural Kenya [51] | Training on how to make a reusable sanitary pad and provision of equipment to make three reusable pads | + | | | | | | | |

**Key**: **0** = No Impact, **+** = Positive Impact

stigma, fear and also increase self-efficacy and promote open discussion about menstruation [31, 41–43, 45, 46, 49, 50, 52]. In Ethiopia, a trial by Blake et al. found that distributing puberty books to schoolgirls improved their menstrual knowledge and attitudes towards menstruation and reduced menstrual shame and fear [45]. The interventions had a significant effect on menstrual hygiene management practices such as increasing bathing during menstruation [31, 43] (Table 2). The impact of menstrual hygiene interventions on lower genital tract infections was assessed in one trial. Though non-significant, there was a lower prevalence of lower genital tract infections among the pooled menstrual cup plus sanitary pad arms, as compared to non-intervention arms [48] (Table 2).

## Discussion

This review included trial studies that implemented different types of menstrual hygiene management interventions using diverse delivery strategies, different types of intervention providers, and different durations of intervention. School attendance, school performance, school dropout, menstrual hygiene knowledge, attitudes, and practices, and aspects of emotional health related to menstruation such as menstrual stigma and shame were outcomes of interest.

Six trial studies indicated a positive effect of interventions on school attendance, school dropout rates, and schoolgirls' academic performance [32, 35, 46, 47, 50, 51]. However, most of the studies had low to moderate levels of bias (Tables 3 and 4). School attendance is usually documented by schoolteachers or by using self-reported diaries, but this method may not provide data accurate enough to reach firm conclusions. In addition, school attendance alone may not be predictive of the academic performance of schoolgirls. Betsu et al. found that a girl's physical presence in a classroom did not necessarily correlate with her mental presence or paying attention [58]. Moreover, mood swings and severe premenstrual symptoms resulting from hormonal changes during menstruation impact paying attention [59]. Additional indicators such as formal educational achievement, school participation, and enrollment in the succeeding grades of school might provide a more robust picture of schoolgirls' academic achievement and continuing access to education. The trial by Nyadoy et al. showed improved academic scores after telling stories and playing games about menstrual hygiene management in the intervention arm; however, the outcome was measured only 6 weeks after the baseline assessment, and the small sample size makes it difficult to draw firm conclusions [46].

On the other hand, some studies have found interventions to have no effect, including the studies that evaluate menstrual cups [34, 43, 48]. Despite being a potentially viable intervention, many people falsely believe that menstrual cups can cause loss of virginity and reduced fertility [60]. Furthermore, menstrual cups may be difficult to clean effectively whenever water supplies are inadequate. This may influence the results of interventions using this device. Studies reporting positive effects, or no intervention effect may suffer from different forms of bias, posing challenges for policymakers and stakeholders looking for evidence-based menstrual hygiene interventions.

The systematic review by Chandra-Mouli et al. highlighted the importance of providing accurate biological information to menstruating girls. In most cases, girls receive most of their information from their mothers, but schoolgirls also seek menstrual information and support from older siblings, and their peers. All of these sources, and particularly their mothers, may be significant sources of menstrual misinformation [17, 58]. There is a great need to improve community knowledge of the biology of menstruation, but despite this fact, mothers as a group are generally not targeted for improved education. In most of the studies we reviewed, premenarchal schoolgirls, in particular, receive insufficient education concerning menstruation, leaving them unprepared for the biological changes associated with menarche and contributing to their frustration, bewilderment, and anxiety when menstruation begins [61].

For schoolgirls to manage menstruation safely and comfortably, they need supportive social norms in the community as well as a welcoming environment both at home and at school. Interventions that improve parental involvement in menstrual hygiene management and that also target community sources of menstrual misinformation are generally lacking in the literature. The study by Sol et al. that engaged parents found a positive effect on menstrual hygiene management and school attendance [50]. Another study by Agbede et al. looking at the combined effect of peer and parent educational interventions had the highest mean score of menstrual hygiene management practice among its study arms [42].

Discussing menstruation with male family members (including, and perhaps especially, fathers) is another challenge faced by schoolgirls [62]. Approximately 13% of Tanzanian girls have encountered period teasing, while over 80% expressed fear of being teased, particularly by male classmates. This results in reduced school attendance, participation, and concentration in class [63]. Another article exploring the beliefs and attitudes boys and men hold about menstruation revealed that men generally have more negative attitudes towards menstruation and view it as debilitating and requiring secrecy. However, these attitudes may soften as men age and gain more knowledge and experience with menstruation [64]. Certain religious texts can be interpreted to associate menstruation with impurity and uncleanliness, which leads to menstrual restrictions, shame, and taboos in some cultures [65]. Many of these challenges are unaddressed by most of the menstrual intervention studies. A qualitative investigation by Betsu et al. indicated that many school teachers support attitudes promoting menstrual secrecy by their comments in class concerning menstrual hygiene, saying things like "drying reusable pads in a hidden place is helpful to prevent *Michi*," a folk-ailment believed by many locals in Ethiopia to be caused by the exposure of used or washed menstrual pads to sunlight [58]. Those who provide menstrual hygiene education and interventions should be aware of common cultural misperceptions regarding menstruation that exist in their communities. Better training of the teachers who provide classroom instruction concerning menstruation is also needed.

The review has certain limitations. Many of the studies that were included in this review relied on self-reported menstrual knowledge, attitude, and practices, as well as potentially inaccurate school attendance records, which could lead to over- or under-estimation of the findings due to poor recording of school attendance, social desirability bias and recall bias. Most of the studies reviewed did not address the impact of water, sanitation, and hygiene interventions or other community-based interventions on menstrual hygiene management. The review consisted of trials that used different interventions and methods of measuring outcomes and included a wide range of ages (9–25 years). These variations make it challenging to compare the studies with one another and to capture accurately the impact of menstrual hygiene interventions on the outcomes of interest. All relevant studies may not have been captured for this review due to limitations in the search strategies and limiting the studies reviewed to those in English. Most of the literature about menstrual hygiene management, especially in low- and middle-income countries, doesn't adequately address the needs of people who identify as gender-nonconforming. Since gender-diverse persons make up such a small percentage of the population, it can be assumed that most of the academic literature on MHM is written from the perspective of cis women. This exclusion may affect menstrual hygiene needs and experiences of transgender, non-binary, and other gender-varied individuals [66]. There was a paucity of randomized controlled trials and quasi-randomized controlled trials, and this may have biased the resulting literature. As a result, conclusions concerning the effectiveness of menstrual hygiene interventions should be interpreted with caution.

The review also has several strengths, providing an extensive summary of English-language evidence. It offers valuable insights by presenting a comprehensive review of English-language

trial studies that evaluate the effect of menstrual hygiene management interventions on various aspects of schoolgirls' lives. With a large sample size from multiple countries, the study covers a broad range of interventions, including puberty education, distribution of menstrual supplies, and integration of menstrual health topics into school curriculums. The findings not only underscore the positive effects of these interventions, such as increased school attendance and enhanced menstrual hygiene knowledge and attitudes but also shed light on the challenges and limitations observed in certain studies. This review will also have a great contribution to identifying research gaps for further studies.

The results of this review have several implications for practice in the field of menstrual hygiene management. It highlights the need for comprehensive and accurate education about menstruation, not only for girls but also for their parents, teachers, and communities. This education should address misconceptions, and cultural taboos, and provide information on appropriate menstrual hygiene practices. This can be implemented by incorporating menstrual hygiene management into school curriculum and training; providing access to affordable and hygienic menstrual products; ensuring adequate water and sanitation facilities and creating supportive environments that reduce stigma and shame associated with menstruation.

The findings of this review have important policy implications. Governments and policy-makers should prioritize menstrual hygiene as a public health issue and develop policies and guidelines to meet the needs of menstruating girls including affordable menstrual products, proper hygiene facilities, education about menstruation, and access to healthcare services for managing menstrual health. It is also crucial to address cultural beliefs and misconceptions surrounding menstruation through awareness campaigns and community engagement. Adequate funding should be allocated to ensure the effective implementation and monitoring of these policies to ensure that all girls have access to menstrual hygiene facilities and education.

While this review provides valuable insights into the effectiveness of menstrual hygiene management interventions, there are several areas that require further research. Future studies should aim to overcome the limitations identified in this review, such as biases and small sample sizes. Longitudinal studies with larger sample sizes are needed to assess the long-term impact of menstrual hygiene management interventions on school attendance, academic performance, and emotional well-being. Additionally, more research is needed to explore the effectiveness of different delivery strategies and intervention providers. It would also be beneficial to investigate the cultural and social factors that influence menstrual hygiene practices and develop interventions that address these specific contexts. Overall, future research should focus on generating robust evidence to inform the development of evidence-based interventions and policies in the field of menstrual hygiene management.

## Conclusion

Menstrual hygiene management interventions can have a positive impact on schoolgirls' attendance, reducing dropout rates, and improving their school performance and emotional health. Moreover, such interventions can improve knowledge, attitudes, and practices pertaining to menstruation and its management. A holistic approach that includes accurate menstrual education, better access to hygiene products, improved water, sanitation, and hygiene facilities; and greater engagement of parents, religious and community leaders is likely to make the greatest impact in this area. Of crucial importance is treating all who menstruate with the respect they deserve and reducing the stigmatization and shame that often surrounds this biological process.

It is also important to standardize the interventions used as well as the tools used to measure outcomes if such programs are to be evaluated properly. This will help to identify best

practices and improve the overall effectiveness of the menstrual hygiene interventions used. Organizing large randomized clinical trials to address these issues in a well-structured manner would be extremely useful in moving our knowledge forward on how to improve menstrual hygiene among schoolgirls in low- and middle-income countries.

## Supporting information

**S1 Checklist. PRISMA checklist.**
(DOCX)

**S1 Fig. Preferred Reporting Items for Systematic Reviews and Meta-analyses (PRISMA) flow chart.**
(TIFF)

**S1 Protocol. For menstrual hygiene management interventions and their effects on schoolgirls' menstrual hygiene experiences in low and middle countries: A systematic review.**
(DOCX)

**S1 Data. Templet data collection form.**
(DOCX)

**S2 Data. Data extracted from the included studies.**
(DOCX)

## Author Contributions

**Conceptualization:** Balem Demtsu Betsu, Araya Abrha Medhanyie, Tesfay Gebregzabher Gebrehiwet, L. Lewis Wall.

**Data curation:** Balem Demtsu Betsu.

**Formal analysis:** Balem Demtsu Betsu.

**Investigation:** Balem Demtsu Betsu.

**Methodology:** Balem Demtsu Betsu.

**Supervision:** Araya Abrha Medhanyie, Tesfay Gebregzabher Gebrehiwet, L. Lewis Wall.

**Validation:** Araya Abrha Medhanyie, Tesfay Gebregzabher Gebrehiwet, L. Lewis Wall.

**Writing – original draft:** Balem Demtsu Betsu.

**Writing – review & editing:** Araya Abrha Medhanyie, Tesfay Gebregzabher Gebrehiwet, L. Lewis Wall.

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
