## [Decision Letter · Decision Letter 0]

5 Dec 2023

PONE-D-23-35026Menstrual hygiene management interventions and their effects on schoolgirls’ menstrual hygiene experiences in low and middle countries: A systematic reviewPLOS ONE

Dear Dr. Betsu,

Thank you for submitting your manuscript to PLOS ONE. After careful consideration, we feel that it has merit but does not fully meet PLOS ONE’s publication criteria as it currently stands. Therefore, we invite you to submit a revised version of the manuscript that addresses the points raised during the review process.

Both reviewers have expressed some minor points which should be easy to address.

We look forward to receiving your revised manuscript.

Kind regards,

Alison Parker

Academic Editor

PLOS ONE

2. PLOS requires an ORCID iD for the corresponding author in Editorial Manager on papers submitted after December 6th, 2016. Please ensure that you have an ORCID iD and that it is validated in Editorial Manager. To do this, go to ‘Update my Information’ (in the upper left-hand corner of the main menu), and click on the Fetch/Validate link next to the ORCID field. This will take you to the ORCID site and allow you to create a new iD or authenticate a pre-existing iD in Editorial Manager. Please see the following video for instructions on linking an ORCID iD to your Editorial Manager account: https://www.youtube.com/watch?v=_xcclfuvtxQ".

3. Please include your tables as part of your main manuscript and remove the individual files. Please note that supplementary tables (should remain/ be uploaded) as separate ""supporting information"" files".

4. Please remove your figures from within your manuscript file, leaving only the individual TIFF/EPS image files, uploaded separately. These will be automatically included in the reviewers’ PDF.".

Reviewers' comments:

Reviewer's Responses to Questions

**Comments to the Author**

1. Is the manuscript technically sound, and do the data support the conclusions?

Reviewer #1: Yes

Reviewer #2: Yes

2. Has the statistical analysis been performed appropriately and rigorously? 

Reviewer #1: Yes

Reviewer #2: N/A

3. Have the authors made all data underlying the findings in their manuscript fully available?

Reviewer #1: Yes

Reviewer #2: Yes

4. Is the manuscript presented in an intelligible fashion and written in standard English?

Reviewer #1: Yes

Reviewer #2: Yes

5. Review Comments to the Author

Reviewer #1: I would encourage the authors to tighten the piece prior to publication in two ways. First, the nature of the search seems to have translated into several important studies escaping capture as part of the dataset. This is not fatal; it is the nature of the searches the authors chose to run. But I think the authors will want to cabin their conclusions as being only as good as the data the searches yielded, acknowledging that other work that did not fit the authors' relatively narrow criteria were therefore excluded. Second, I think the authors would be well-served to address the linguistic challenges in talking about "women and girls" and "menstrual hygiene," when those terms have been made much more complicated by excellent work in this field. The authors gesture at this by the use of the term "menstruand" (which I found jarring and out of synch with most other scholarship in the field," but they never address the issue head-on.

Reviewer #2: General

Much is written in passive voice – active voice would be preferable. Occasionally you use ‘we’ – you should pick one narrative style and stick to it.

I don’t think it’s necessary to have the acronyms listed since there are only two (unless this is a requirement of the journal then please ignore me). You haven’t included PICOT.

There are times when you spell out MHM and others when you use the acronym – be consistent and use the acronym throughout

The paper should recognise that it isn’t just cisgender women who menstruate but potentially also transgender men, non-binary and other gender diverse persons. The term ‘menstruator’ could be used instead of women and girls to be inclusive.

I think it would be helpful if you stated the countries where studies present different evidence else you might fall into making sweeping generalising comments. For example lines 88-89.

You’ve used WASH as an acronym and spelled out but haven’t introduced it as an acronym – please revise

You could consider adding a positionality statement at the beginning of the paper.

Why were no studies in any Ethiopian languages selected?

Abstract

‘intervention’ seems like a rather broad keyword

Introduction

MHM definition – could point out that depending on the materials used somewhere to wash and dry reusable materials is also necessary. Could also mention the access to and ability to wash and dry underwear is also a necessity.

Lines 84-87 its clearer written as: More than half (52 %) of adolescent girls in Ethiopia

have never received any information about menstrual hygiene due to socio-cultural

87 misinformation, religious taboos, and inadequate menstrual supplies and facilities, which leads to fear, confusion, and lack of confidence when menarche occurs (11-15).

Line 93 – what is meant by gender empowerment exactly? Vague phrase it might be interesting to unpack it.

111-112 you use the word review a lot

Good justification for the work.

Methods

I wonder if it would produce, more results if you googled gender neutral terms like ‘menstruator’ – it could be written as a limitation if you didn’t do this.

The systematic review following PRISMA is clearly and transparently described. Having three reviewers to judge papers against the outlined eligibility criteria reduces bias.

Line 175 - I don’t think mode of intervention needs an explanation.

The search strategy is well documented and comprehensive

Data bases and other sources of information are specified

The process of study selection is clearly outlined.

The process of data extraction is well documented

The key characteristics of included studies is clearly presented.

The studies were rigorously assessed for risk of bias using the Joanna Briggs Institute critical appraisal assessment tools

Results

Line 192-193 - I think you mean Saudi Arabia, not Saudi Riyadh

From like 251 – have you listed all of the education components because saying they ‘included things such as…’ makes it seem that the list is not exhaustive but it seems to be that way. And perhaps it should be if not too long to include each component.

Line 279 - Good and transparent analysis of intervention fidelity

Interesting discussion on blinding.

Line 305 – et al is repeated

Table 2 and 3 – could we have colour code for severity of bias?

Table 4 – reduce space between lines so table is less spread out – its currently across 13 pages

Table 4 – expand acronyms

Table 5 – I assume the 0 means no impact and + means positive impact but you need to dd this key somewhere. It would be clearer if you also did a colour i.e. green for positive effect , yellow for neutral. It would also look better if the symbols were in the centre of the square. Table 5 – could you add the type of intervention into the summary table and group similar types of intervention together?

Discussion

370 – was the not paying attention due to hormonal fluctuations or worry and concern about not being able to manage their period?

An important narrative I feel is missing more on which there is a growing body of literature on is the role of men/boys/non-menstruators is upholding stigma/teasing etc. you touch upon this in lines 405-406 but feel some more literature on how non-menstruators act as a barrier could be added in.

407 – careful not to generalise all religious beliefs be specific.

Implications

447-449 – be more specific – state the unmet needs

Conclusion

Seems like the conclusion is the first instance apart from the title where you mention low and middle income countries… why would this not be an issue in high income countries? The United Kingdom is currently going through a cost of living crises where people are in hygiene poverty – unable to buy basic hygiene items like menstrual pads. Ref - https://journals.plos.org/plosone/article?id=10.1371/journal.pone.0255001&utm_source=miragenews&utm_medium=miragenews&utm_campaign=news

Furthermore – Saudi Arabia is one of the countries studied, which is a high income country.

Is categorising countries by their economic status quite western and neoliberal?

6. PLOS authors have the option to publish the peer review history of their article (what does this mean?). If published, this will include your full peer review and any attached files.

Reviewer #1: No

Reviewer #2: **Yes: **Georgia Hales

---

## [Author Response · Author response to Decision Letter 0]

17 Jan 2024

To Reviewer 1

• Dear reviewer, the conclusion provided in the manuscript is yielded from the findings (literatures reviewed). The findings evidenced that the interventions can enhance schoolgirls' educational outcomes, and can improve their menstrual knowledge, attitudes, and practices by helping them manage their periods more effectively. 

• We used the term “Menstruant” not “menstruand” may be it is editorial error. The use of women and girl is more dominant in WASH sectors. While menstruants is emerging form gender inclusive perspective, hence, in the write up we tried to balance the use of the terms. However our search clearly identified literatures that assessed the effect of menstrual interventions on “school girls”. 

To Reviewer 2

Authors’ response

• We have converted passive voice to active voice in most of the relevant sections.

• Acronyms have been spelled out and made consistent in the current version.

• Although "Menstruants" is a gender inclusive term, the primary objective is to assess its effect on “school girls” and we had to be specific to these group of study participants. Moreover, the studies we have reviewed mostly used the term "Schoolgirls," "Girls," and "Women". For this reason, we maintained the terms. However, in the write up section we have also considered “menstruants” 

• The findings are form different low and middle income countries, and citation is in place ( line 89)

• The academic and research language for Ethiopia is English. So, there is no study conducted using Ethiopian language and which is not included. 

• We have included positionality statement under the method section on the current version (line 128)

Abstract

‘intervention’ seems like a rather broad keyword

Authors’ response

 “Intervention" is preceded by "menstrual hygiene management" and expressed as “menstrual hygiene management interventions” to be more specific, (lines 27 and 34)

MHM definition – could point out that depending on the materials used somewhere to wash and dry reusable materials is also necessary. Could also mention the access to and ability to wash and dry underwear is also a necessity.

Authors’ Response: Agreed, we utilized the standard definitions, and employing operational definitions may provide further assistance. 

Lines 84-87 its clearer written as: More than half (52%) of adolescent girls in Ethiopia have never received any information about menstrual hygiene due to socio-cultural

87 misinformation, religious taboos, and inadequate menstrual supplies and facilities, which leads to fear, confusion, and lack of confidence when menarche occurs (11-15).

Line 93 – what is meant by gender empowerment exactly? Vague phrase it might be interesting to unpack it.

111-112 you use the word review a lot

Good justification for the work.

 Authors’ response 

The issue is fixed accordingly: (line 86-89)

• “More than half (52 %) of adolescent girls in Ethiopia have never received any information about menstrual hygiene (10), due to religious taboos, socio-cultural misinformation, and inadequate menstrual supplies and facilities, which leads to fear, confusion, and lack of confidence when menarche occurs (11-15).” 

• “gender empowerment” modified to “gender equality” ( line 93)

Methods

I wonder if it would produce, more results if you googled gender neutral terms like ‘menstruator’ – it could be written as a limitation if you didn’t do this.

Authors’ response 

• In fact, we purposefully limited our search terms to be gender-specific to highlight the impact the interventions on schoolgirls, but it is still valued concern because it may have limited searching relevant literature. We have included it on the limitation section of the manuscript 

The systematic review following PRISMA is clearly and transparently described. Having three reviewers to judge papers against the outlined eligibility criteria reduces bias.

Line 175 - I don’t think mode of intervention needs an explanation.

The search strategy is well documented and comprehensive

Data bases and other sources of information are specified

The process of study selection is clearly outlined.

The process of data extraction is well documented

The key characteristics of included studies is clearly presented.

The studies were rigorously assessed for risk of bias using the Joanna Briggs Institute critical appraisal assessment tools

Authors’ Response

Written as “Mode of intervention” only (Line 200, in the modified version) 

Results

Line 192-193 - I think you mean Saudi Arabia, not Saudi Riyadh 

Authors’ Response 

• Saudi Arabiya is excluded from the analysis as it is one of the high income countries. And modification is made on the manuscript accordingly ( Line 215)

From line 251 – have you listed all of the education components because saying they ‘included things such as…’ makes it seem that the list is not exhaustive, but it seems to be that way. And perhaps it should be if not too long to include each component.

Authors’ Response 

 Modified as “The menstrual education components of the studies included; puberty education, training on…” (line 239)

Line 279 - Good and transparent analysis of intervention fidelity

Interesting discussion on blinding.

Line 305 – et al is repeated

Authors’ response: Amendment made (the repeated et al is deleted/ line 295) 

Table 2 and 3 – could we have color code for severity of bias?

 Authors’ response: Color code is given accordingly 

Table 4 – reduce space between lines so table is less spread out – its currently across 13 pages

Authors’ response: Resolved (line space of the table is reduced) 

Table 4 – expand acronyms

Table 5 – I assume the 0 means no impact and + means positive impact but you need to add this key somewhere. It would be clearer if you also did a colour i.e. green for positive effect, yellow for neutral. It would also look better if the symbols were in the centre of the square. Table 5 – could you add the type of intervention into the summary table and group similar types of intervention together?

Authors’ response

• Acronyms are expand on the heading section

• Key is provided for Table-4 on line 337

 Discussion

370 – was the not paying attention due to hormonal fluctuations or worry and concern about not being able to manage their period?

Authors’ response: lack of attention in education could be attributed by both the hormonal fluctuations and worry about not being able to manage their period. We have added additional literatures to illustrate it. (Line 353)

An important narrative I feel is missing more on which there is a growing body of literature on is the role of men/boys/non-menstruators is upholding stigma/teasing etc. you touch upon this in lines 405-406 but feel some more literature on how non-menstruators act as a barrier could be added in.

Authors’ Response: Noted and we have added more details as follows 

“One article exploring the beliefs and attitudes boys and men hold about menstruation revealed that men generally have more negative attitudes towards menstruation and view it as debilitating and requiring secrecy. However, these attitudes may soften as men age and gain more knowledge and experience with menstruation” (line 390)

407 – careful not to generalize all religious beliefs be specific.

Authors’ response: Addressed in line 394 as “most religious beliefs”, to avoid generalization.

Implications

447-449 – be more specific – state the unmet needs

Authors’ Response: specified on the current version on line 435-439

“Governments and policymakers should prioritize menstrual hygiene as a public health issue and develop policies and guidelines to meet the needs of menstruating girls including affordable menstrual products, proper hygiene facilities, education about menstruation, and access to healthcare services for managing menstrual health.”

Conclusion

Seems like the conclusion is the first instance apart from the title where you mention low- and middle-income countries… why would this not be an issue in high income countries? The United Kingdom is currently going through a cost-of-living crises where people are in hygiene poverty – unable to buy basic hygiene items like menstrual pads. Ref - https://journals.plos.org/plosone/article?id=10.1371/journal.pone.0255001&utm_source=miragenews&utm_medium=miragenews&utm_campaign=news

Furthermore – Saudi Arabia is one of the countries studied, which is a high income country. 

Is categorizing countries by their economic status quite western and neoliberal?

Authors’ response 

Menstrual hygiene management is indeed an issue that affects individuals in both high-income and low-income countries. In high-income countries, access to menstrual products and facilities for proper hygiene may be more readily available (as compared to low income countries), but issues such as stigma and access to education about menstrual hygiene persist. In low-income countries, the challenges may include limited access to sanitary products, clean water, and sanitation facilities, as well as social stigma and inadequate education about menstrual health. Therefore, addressing menstrual hygiene management is an important aspect of promoting gender equality and ensuring the well-being of individuals across different socioeconomic contexts.

We have used the World Bank classification of countries based on economic status and we have excluded the finding from Saudi Arabia. 

Categorization is made by income level ( world bank)

---

## [Decision Letter · Decision Letter 1]

29 Jan 2024

PONE-D-23-35026R1Menstrual hygiene management interventions and their effects on schoolgirls’ menstrual hygiene experiences in low and middle countries: A systematic reviewPLOS ONE

Dear Dr. Betsu,

Thank you for submitting your manuscript to PLOS ONE. After careful consideration, we feel that it has merit but does not fully meet PLOS ONE’s publication criteria as it currently stands. Therefore, we invite you to submit a revised version of the manuscript that addresses the points raised during the review process.

Thanks for doing a good job with the first round of comments.   I'm afraid both reviewers have some further things for you to address but I think these should be pretty straightforward now. Please submit your revised manuscript by Mar 14 2024 11:59PM. If you will need more time than this to complete your revisions, please reply to this message or contact the journal office at plosone@plos.org. Please include the following items when submitting your revised manuscript:A rebuttal letter that responds to each point raised by the academic editor and reviewer(s). You should upload this letter as a separate file labeled 'Response to Reviewers'.A marked-up copy of your manuscript that highlights changes made to the original version. You should upload this as a separate file labeled 'Revised Manuscript with Track Changes'.An unmarked version of your revised paper without tracked changes. You should upload this as a separate file labeled 'Manuscript'.If applicable, we recommend that you deposit your laboratory protocols in protocols.io to enhance the reproducibility of your results. Protocols.io assigns your protocol its own identifier (DOI) so that it can be cited independently in the future. For instructions see: https://journals.plos.org/plosone/s/submission-guidelines#loc-laboratory-protocols. Additionally, PLOS ONE offers an option for publishing peer-reviewed Lab Protocol articles, which describe protocols hosted on protocols.io. Read more information on sharing protocols at https://plos.org/protocols?utm_medium=editorial-email&utm_source=authorletters&utm_campaign=protocols.

We look forward to receiving your revised manuscript.

Kind regards,

Alison Parker

Academic Editor

PLOS ONE

Journal Requirements:

Reviewers' comments:

Reviewer's Responses to Questions

**Comments to the Author**

1. If the authors have adequately addressed your comments raised in a previous round of review and you feel that this manuscript is now acceptable for publication, you may indicate that here to bypass the “Comments to the Author” section, enter your conflict of interest statement in the “Confidential to Editor” section, and submit your "Accept" recommendation.

Reviewer #1: (No Response)

Reviewer #2: (No Response)

2. Is the manuscript technically sound, and do the data support the conclusions?

Reviewer #1: Yes

Reviewer #2: Yes

3. Has the statistical analysis been performed appropriately and rigorously? 

Reviewer #1: Yes

Reviewer #2: Yes

4. Have the authors made all data underlying the findings in their manuscript fully available?

Reviewer #1: Yes

Reviewer #2: Yes

5. Is the manuscript presented in an intelligible fashion and written in standard English?

Reviewer #1: Yes

Reviewer #2: Yes

6. Review Comments to the Author

Reviewer #1: Line 24: uses passive voice

Line 32: “Accordingly” doesn’t flow from previous sentence

Line 35: “However” is clunky here.

Line 82: You use the word “menstruants” without explaining this word choice. This absolutely needs to be addressed. It is not widely accepted (by two different “camps” for different reasons). It is jarring and merits context. If you believe that this is the term that is emerging, you should at least cite to some authority for that, because I don’t think there is consensus/agreement that many readers will have heard it before (“menstruator” is much more common in significant literature).

Lines 84-89: You definitely need to explain your focus on women and girls. Your response to the reviewers makes clear that it is because that is what the studies you survey are studying…but this does not come through in your paper. The reader is still left wondering why you are focusing on *girls* (but then again, are you focused just on girls? In fact, later in the paper, such as at Lines 296-98, you do cite studies involving boys). This needs to be clarified.

Lines 95-96: I don’t think the average reader will understand what you mean by “weak enabling environments”’

Line 111: comma missing

Line 114: reviews “were conducted” (passive voice) by WHOM?

Lines 143-145: Here you talk about limiting your search to studies in which the participants were schoolgirls only. Is this accurate? See comment above re Lined 296-98. Perhaps I am misunderstanding.

Line 148: Punctuation missing?

Lines 149, 152, 160: Colon not necessary?

Line 165: Unnecessary comma

Line 210: Do you mean “findings” (plural)?

Line 212: Are “menstrual interventions” the same as “menstrual hygiene management interventions” discussed on Line 146?

Line 254: Passive voice

Line 258: Odd initial cap for “School”

Lines 272-73: Awkward and difficult to follow sentence.

Line 298: Were these period underwear or regular underwear?

Lines 300-302: Run-on sentence.

Line 464: Instead of talking about “menstruating girls and women,” might this be an appropriate place to talk about “all who menstruate,” notwithstanding your focus?

Line 466: Queer theory would ask us to look rigorously at the word “normal.” I think you mean “involuntary” or “inevitable” (or perhaps drop the adjective entirely).

Reviewer #2: Dear authors,

Thanks for the effort you’ve put in to addressing my comments. I feel like the paper has now avoided a couple of easy pitfalls. The information in the tables is also now much easier for the reader to digest – thanks for this. There are just a few things I’m not sure were fully addressed; perhaps I did not explain myself very well so here I elucidate:

I understand the want to just focus on one particular group (schoolgirls). However, I raised this point as it is often gender diverse persons who are left out and not represented in this type of data collection, which is something that needs to change if the research is to remain relevant going forwards. I appreciate the mention of menstruators – it’s important as a limitation if anything to explain the data set is solely focused on cisgender girls and women.

Largely you give the country where the reference is from but not all the time. It’s important to do this consistently in order to avoid making generalisation about all low-income countries.

I’m happy to see the inclusion of the positionality statement – it adds transparency. Something I was looking for was mention of the country/countries the authors are from. There adds incongruence if the authors were from a high-income country say but are commenting on low income countries. How does where the authors are from impact the direction of the study or interpretation of results?

That’s a shame that you’ve now had to exclude Saudi Arabia. I hope my point that menstrual health is a global issue that impacts high-income countries was taken not to diminish the lack of access in lower income countries but to highlight it’s not just an issue in low-income countries and that there are inequalities within countries as well as between them.

Great that you’ve added in some literature regarding men/boys/non-menstruators however the point that I wanted to get at was that it can be these groups that also act as a barrier to schoolgirls’ attendance through teasing or shaming.

I don’t think changing the line to ‘most religious beliefs’ avoids generalisation either… this is a sensitive point and I understand what you’re getting at… but I think what you’re saying could be misinterpreted. Religion is interpreted and enacted differently across the world. For example in the UK, I don’t think many Christians would feel stigma towards menstruation because of what is written in the Bible. But I understand that other cultures might. What you can say instead which is factually true is that in certain religious texts menstruation is presented as making the menstruator impure or unclean. I would be particular about using the exact wording – so much of religious text is down to interpretation.

I feel that for this work to be relevant, contemporary, and self-aware, the authors should take a little more time to contemplate these last points. It would also be good to have a statement from the authors on why this paper is important and what new contribution to the discussions on MHM it brings.

Many thanks,

Reviewer 2

7. PLOS authors have the option to publish the peer review history of their article (what does this mean?). If published, this will include your full peer review and any attached files.

Reviewer #1: No

Reviewer #2: **Yes: **Georgia Hales

---

## [Author Response · Author response to Decision Letter 1]

4 Mar 2024

Reviewer #1: 

Line 24: uses passive voice

 Authors ‘Response 

Dear reviewers I agree with comments in line #24 

The statement in line# 24 is replaced with active voice, “To address these issues, researchers have conducted intervention studies, but the impact on school attendance has varied”

Line 32: “Accordingly” doesn’t flow from previous sentence

Authors ‘Response 

I agree with comment of the reviewer, the statement in line 32 is paraphrased as 

“Review of sixteen trial studies showed that menstrual hygiene interventions have a positive effect on schoolgirls' school attendance, performance, and dropout rates, as well as on their menstrual knowledge, attitudes, practices, and emotional well-being.”

Line 35: “However” is clunky here.

Authors ‘Response 

The comment is well taken and “However” in line # 34 is omitted and the sentences is rephrased as “There was a low to medium risk of bias in the most of the studies.”

Line 82: You use the word “menstruants” without explaining this word choice. This absolutely needs to be addressed. It is not widely accepted (by two different “camps” for different reasons). It is jarring and merits context. If you believe that this is the term that is emerging, you should at least cite to some authority for that, because I don’t think there is consensus/agreement that many readers will have heard it before (“menstruator” is much more common in significant literature).

Authors ‘Response

I agree with point raised and it is valued concern. Hence the word “menstruants” in line 82 is replaced with “menstruators” as per the recommendation 

Lines 84-89: You definitely need to explain your focus on women and girls. Your response to the reviewers makes clear that it is because that is what the studies you survey are studying…but this does not come through in your paper. The reader is still left wondering why you are focusing on *girls* (but then again, are you focused just on girls? In fact, later in the paper, such as at Lines 296-98, you do cite studies involving boys). This needs to be clarified.

Authors ‘Response 

The comment is well accepted. The reason for focusing on schoolgirls is now explained in line 419-426 as follows, to make it clear to the reader.

“Most of the literature about menstrual hygiene management, especially in low- and middle-income counties, don’t adequately address the needs of people who identify as gender-nonconforming. Menstrual discourses, are frequently written from the perspective of a cis woman, highlighting only the menstrual experiences of adolescent girls and cis women. This exclusion may affect menstrual hygiene needs and experiences of transgender, non-binary, and other gender varied individuals”

This review investigated how involving males (boys, fathers, or parents) in interventions might improve schoolgirls' experiences. We included intervention studies targeting these male groups alongside traditional MHM interventions. This broadens the range of interventions studied, but the outcome/impact must be measured specifically on schoolgirls.

Lines 95-96: I don’t think the average reader will understand what you mean by “weak enabling environments”’

Authors ‘Response 

The comment is well accepted. In the old version we explained the word “weak enabling environments”’ using the list of challenges that came after it . i.e. “Insufficient knowledge about menstruation, inadequate access to water, sanitation and hygiene services, lack of adequate hygiene materials, and social norms unsupportive of those who menstruate” 

But if it creates confusion for readers, the already listed challenges are more descriptive and we have omitted the phrase “weak enabling environment” to avoid ambiguity to the reader

Line 111: comma missing

Authors ‘Response . 

This is well taken and Comma is used in line #110 “2015-2016 school year, demonstrated”

Line 114: reviews “were conducted” (passive voice) by WHOM?

Authors ‘Response 

Dear reviewer, we have included the list of 4 citations which helps to address the potential question about “who conducted the review?”. The 4 citations in line 116 can address this concern than listing the name of the authors who did the review. 

Lines 143-145: Here you talk about limiting your search to studies in which the participants were schoolgirls only. Is this accurate? See comment above re Lined 296-98. Perhaps I am misunderstanding.

Authors ‘Response 

Dear reviewer, your understanding is correct. The statement is rephrased to “

 The search was limited to studies that measured outcomes on schoolgirls because the objective of the review was to evaluate how menstrual hygiene management intervention programs impact schoolgirls' attendance, academic performance, or dropout rates. ” The study aimed to evaluate the effects of intervention programs on school attendance, performance, and dropout rates of schoolgirls, as clearly stated in lines 144-145. Therefore, while the intervention may involve parents, the community, and males, the outcomes must be assessed specifically for schoolgirls. We have included studies that involved parents or males in the intervention, with the actual outcomes measured for the schoolgirls. Additionally, we used terms such as fathers, mothers, community, and parents as search terms to encompass various types of interventions.

Line 148: Punctuation missing?

Authors ‘Response 

This is well noted punctuation is in place now, “supplies.”

Lines 149, 152, 160: Colon not necessary?

Authors ‘Response 

Well noted and accepted, the colon in line 149 and 152 is removed and we kept the colon in line 159 and we deleted the word “includes” to make appropriate use of colon 

 Line 165: Unnecessary comma

Authors ‘Response 

Well accepted, comma in line 165 is removed the statement is written as “we excluded studies not available in the English language and conference abstracts.”

Line 210: Do you mean “findings” (plural)?

Authors ‘Response 

This is noted, line 208 in this version, is changed into plural “a summary of the findings”

Line 212: Are “menstrual interventions” the same as “menstrual hygiene management interventions” discussed on Line 146?

Authors ‘Response 

Yes, it is the same and line 210 is corrected as “Sixteen trial studies that assessed the effect of menstrual hygiene management interventions…”

Line 254: Passive voice

Authors ‘Response 

Dear reviewer the comment is accepted, the statement in line 252 is changed into active voice as follows 

 “Montgomery et al. suggested that using school attendance and dropout rates as a proxy indicator of academic performance”

Line 258: Odd initial cap for “School”

Authors ‘Response 

This is accepted. Line 256 Changed from Official School attendance� Official school attendance

Lines 272-73: Awkward and difficult to follow sentence.

Authors ‘Response 

The comment is well accepted. The statement in line 270-272 which was “Montgomery et al. conducted study on comparable peri-urban schools but included one remote rural site without experience in using sanitary pads that had no electricity, and no paved roads.”is rephrased as follows: 

“Montgomery et al. carried out study on peri-urban schools that were comparable, but they also incorporated a remote rural site lacking experience in using sanitary pads, with no access to electricity and unpaved roads.” 

Line 298: Were these period underwear or regular underwear?

Authors ‘Response 

Dear reviewer, fortunately I physically knew the underwear’s provided in this specific intervention and they are normal/regular underwear’s and the term “ regular” is added ( line 297-298)

Lines 300-302: Run-on sentence.

Authors ‘Response 

Well noted and accepted. The statement in line 297-299 “School attendance (data collected prospectively on-site by team researchers) was analyzed before and after the intervention and also compared with attendance data from the prior school year”

Is amended as 

“School attendance was analyzed before and after the intervention and also compared with attendance data from the prior school year”

Line 464: Instead of talking about “menstruating girls and women,” might this be an appropriate place to talk about “all who menstruate,” notwithstanding your focus

Authors ‘Response 

Dear reviewer this comment is accepted. “Menstruating girls and women “is replaced with “all who menstruate” on the current version ( line # 470) 

Line 466: Queer theory would ask us to look rigorously at the word “normal.” I think you mean “involuntary” or “inevitable” (or perhaps drop the adjective entirely) (line # 474).

Authors ‘Response 

Well Noted, the word “Normal” is dropped from line # 471

Reviewer #2: Dear authors,

Thanks for the effort you’ve put in to addressing my comments. I feel like the paper has now avoided a couple of easy pitfalls. The information in the tables is also now much easier for the reader to digest – thanks for this. There are just a few things I’m not sure were fully addressed; perhaps I did not explain myself very well so here I elucidate:

I understand the want to just focus on one particular group (schoolgirls). However, I raised this point as it is often gender diverse persons who are left out and not represented in this type of data collection, which is something that needs to change if the research is to remain relevant going forwards. I appreciate the mention of menstruators – it’s important as a limitation if anything to explain the data set is solely focused on cisgender girls and women.

Authors ‘Response 

The following paragraph would explain the limitation, which is placed in line # 416-421

“Most of the literature about menstrual hygiene management, especially in low- and middle-income counties, don’t adequately address the needs of people who identify as gender-nonconforming. Menstrual hygiene issues, are frequently written from the perspective of a cis woman, highlighting only the menstrual experiences of adolescent girls and cis women. This exclusion may affect menstrual hygiene needs and experiences of transgender, non-binary, and other gender varied individuals.”

Largely you give the country where the reference is from but not all the time. It’s important to do this consistently in order to avoid making generalization about all low-income countries.

Authors ‘Response 

The input is considered and included the countries where the citation exists, but country was not included mainly on table 5

I’m happy to see the inclusion of the positionality statement – it adds transparency. Something I was looking for was mention of the country/countries the authors are from. There adds incongruence if the authors were from a high-income country say but are commenting on low income countries. How does where the authors are from impact the direction of the study or interpretation of results?

Authors ‘Response 

Dear reviewer this is well accepted. The following statement is added the statement “I am from Ethiopia one of the low-income countries” in line 130 to indicate the principal investigator is from low income country. 

That’s a shame that you’ve now had to exclude Saudi Arabia. I hope my point that menstrual health is a global issue that impacts high-income countries was taken not to diminish the lack of access in lower income countries but to highlight it’s not just an issue in low-income countries and that there are inequalities within countries as well as between them.

Authors ‘Response 

 Dear reviewer, issues of menstrual poverty and inequity are global challenges. It is widely recognized that individuals who menstruate in high-income countries encounter various challenges, although the severity and nature of these challenges may vary from LMIC. Our initial focus for the review was on addressing issues in low and middle-income countries. And we wrongly classified Saudi Arabia as LMIC and include it in the review process. However, this does not imply that menstrual hygiene management is not an issue in high income countries. Your feedback in the previous review highlighted the inclusion of one high-income country, Saudi Arabia, in the review. This was an important observation. It would not be appropriate to include a high-income country when the review is intended to focus on LMICs. This was the sole reason for its exclusion.

Great that you’ve added in some literature regarding men/boys/non-menstruators however the point that I wanted to get at was that it can be these groups that also act as a barrier to schoolgirls’ attendance through teasing or shaming.

Authors ‘Response 

 Dear reviewer the comment is well received. The following statement is included in line #389-392 on the current version which may address the concern about the effect of teasing by boys on school attendance and participation. 

“Approximately 13% of Tanzanian girls have encountered period teasing, while over 80% expressed fear of being teased, particularly by male classmates. This results in reduced school attendance, participation, and concertation in class.” 

I don’t think changing the line to ‘most religious beliefs’ avoids generalization either… this is a sensitive point or I understand what you’re getting at… but I think what you’re saying could be misinterpreted. Religion is interpreted and enacted differently across the world. For example in the UK, I don’t think many Christians would feel stigma towards menstruation because of what is written in the Bible. But I understand that other cultures might. What you can say instead which is factually true is that in certain religious texts menstruation is presented as making the menstruator impure or unclean. I would be particular about using the exact wording – so much of religious text is down to interpretation.

Authors ‘Response 

This is well accepted and the statement “Menstrual restrictions and cultural taboos are often rooted in most religious beliefs and untrue cultural assumptions” Is replaced with following statement on the current version (line #395-397) 

“In certain religious texts, menstruation is often framed as making the menstruator impure or unclean, leading to menstrual restrictions, shame, and taboos”

I feel that for this work to be relevant, contemporary, and self-aware, the authors should take a little more time to contemplate these last points. It would also be good to have a statement from the authors on why this paper is important and what new contribution to the discussions on MHM it brings.

 Authors ‘Response 

The contribution of the study to the discussion on MHM is highlighted as follows in the strength section (line #425-434)

The review also has several strengths, providing an extensive summary of English-language evidence. It offers valuable insights by presenting a comprehensive review of English-language trial studies that evaluate the effect of menstrual hygiene management interventions on various aspects of schoolgirls' lives. With a large sample size from multiple countries, the study covers a broad range of interventions, including puberty education, distribution of menstrual supplies, and integration of menstrual health topics into school curriculums. The findings not only underscore the positive effects of these interventions, such as increased school attendance and enhanced menstrual hygiene knowledge and attitudes, but also shed light on the challenges and limitations observed in certain studies. This review will also have great contribution in identifying research gaps for further studies.

---

## [Decision Letter · Decision Letter 2]

18 Mar 2024

PONE-D-23-35026R2Menstrual hygiene management interventions and their effects on schoolgirls’ menstrual hygiene experiences in low and middle countries: A systematic reviewPLOS ONE

Dear Dr. Betsu,

Thank you for submitting your manuscript to PLOS ONE. After careful consideration, we feel that it has merit but does not fully meet PLOS ONE’s publication criteria as it currently stands. Therefore, we invite you to submit a revised version of the manuscript that addresses the points raised during the review process. **There are just a few more minor points to address, as well as the need to check the grammar further.**

We look forward to receiving your revised manuscript.

Kind regards,

Alison Parker

Academic Editor

PLOS ONE

Journal Requirements:

Reviewers' comments:

Reviewer's Responses to Questions

**Comments to the Author**

1. If the authors have adequately addressed your comments raised in a previous round of review and you feel that this manuscript is now acceptable for publication, you may indicate that here to bypass the “Comments to the Author” section, enter your conflict of interest statement in the “Confidential to Editor” section, and submit your "Accept" recommendation.

Reviewer #1: All comments have been addressed

Reviewer #2: (No Response)

2. Is the manuscript technically sound, and do the data support the conclusions?

Reviewer #1: Yes

Reviewer #2: Yes

3. Has the statistical analysis been performed appropriately and rigorously? 

Reviewer #1: N/A

Reviewer #2: Yes

4. Have the authors made all data underlying the findings in their manuscript fully available?

Reviewer #1: Yes

Reviewer #2: Yes

5. Is the manuscript presented in an intelligible fashion and written in standard English?

Reviewer #1: Yes

Reviewer #2: Yes

6. Review Comments to the Author

Reviewer #1: I implore the authors to run their manuscript through a grammar-checking program (Grammarly, Paperpal, anything) or Chat GPT to improve the grammar and punctuation, especially in the first three pages. The authors have nicely addressed the substantive comments and have responded to specific grammatical errors pointed out by the reviewers. The manuscript still needs to be gone through line by line because parts of it are not well-edited. I do not consider it a good use of my professional time to provide a third round of input pointing out grammatical errors that could be addressed readily by the authors.

Reviewer #2: Thanks for addressing the comments. There are just a couple more tiny things to address and then I’m happy for the work to be published.

1. ‘frequently written from the perspective of a cis woman’ – although I completely agree with this perhaps it requires a reference? Maybe you could explain why you know this to be true as I’m not sure you’d actually be able to find a reference. You could say ‘Since gender-diverse persons make up such a small percentage of the population, it can be assumed that the vast majority of academic literature on MHM is written from the perspective of cis women’. Maybe I’m being unnecessarily pedantic here.

2. I think you would do well to read around the purpose of positionality statements and this would help to inform how to write one. I wasn’t looking for you just to state that you are Ethiopian but how that background and identity shapes and influences the research. For example something like: I am a woman (?) who comes from Ethiopia, which is one of the countries studied in this paper. This gives me first-hand experience of what it’s like to be a menstruating schoolgirl in an LMIC. However, I am an outsider to the other countries studied in this paper, which may leave room for bias or misunderstandings in the interpretation of results’. Something like this.

3. Perhaps I’m being too cautious but I still believe this could be interpreted as offensive to some readers ‘In certain religious texts, menstruation is often framed as making the menstruator impure or unclean, leading to menstrual restrictions, shame, and taboos’. To reiterate, so much religious text is down to interpretation and this is what can cause the issues of negative views towards menstruation, not necessarily the words in the text itself. I don’t know if you’re religious or not but this would be an instance where your positionality affects how you interpret data. For example, I’m not religious and therefore don’t understand or have association to any religious texts. I gave a lecture on MHM making the exact same point as you that religious texts e.g. The Quran says that menstruation is impure. This was of course from a translation from Arabic into English where meaning can get lost anyway. A Muslim student put their hand up to explain that this was a very negative interpretation of the text and that it doesn’t mean impure in a dirty way as was interpreted. I think it would be inoffensive but still make the same point to say ‘certain religious texts can be interpreted to associate menstruation with impurity and uncleanliness, which leads to menstrual restrictions, shame, and taboos in some cultures’ you need a reference for this as well.

7. PLOS authors have the option to publish the peer review history of their article (what does this mean?). If published, this will include your full peer review and any attached files.

Reviewer #1: No

Reviewer #2: **Yes: **Georgia Hales

---

## [Author Response · Author response to Decision Letter 2]

31 Mar 2024

Response to Reviewers

1. If the authors have adequately addressed your comments raised in a previous round of review and you feel that this manuscript is now acceptable for publication, you may indicate that here to bypass the “Comments to the Author” section, enter your conflict-of-interest statement in the “Confidential to Editor” section, and submit your "Accept" recommendation.

Reviewer #1: All comments have been addressed

Reviewer #2: (No Response)

2. Is the manuscript technically sound, and do the data support the conclusions?

Reviewer #1: Yes

Reviewer #2: Yes

3. Has the statistical analysis been performed appropriately and rigorously?

Reviewer #1: N/A

Reviewer #2: Yes

4. Have the authors made all data underlying the findings in their manuscript fully available?

Reviewer #1: Yes

Reviewer #2: Yes

5. Is the manuscript presented in an intelligible fashion and written in standard English?

Reviewer #1: Yes

Reviewer #2: Yes

6. Review Comments to the Author

 Reviewer #1: I implore the authors to run their manuscript through a grammar-checking program (Grammarly, Paperpal, anything) or Chat GPT to improve the grammar and punctuation, especially in the first three pages. The authors have nicely addressed the substantive comments and have responded to specific grammatical errors pointed out by the reviewers. The manuscript still needs to be gone through line by line because parts of it are not well-edited. I do not consider it a good use of my professional time to provide a third round of input pointing out grammatical errors that could be addressed readily by the authors.

Authors’ Response:

Dear reviewer, this is valued concern. The necessary grammar and spelling errors are fixed throughout the document on the current version manuscript. 

Reviewer #2: Thanks for addressing the comments. There are just a couple tinier things to address and then I’m happy for the work to be published.

1. ‘frequently written from the perspective of a cis woman’ – although I completely agree with this perhaps it requires a reference? Maybe you could explain why you know this to be true as I’m not sure you’d actually be able to find a reference. You could say ‘Since gender-diverse persons make up such a small percentage of the population, it can be assumed that the vast majority of academic literature on MHM is written from the perspective of cis women’. Maybe I’m being unnecessarily pedantic here.

 Authors’ Response:

Dear reviewer this comment is valued and is addressed as follows in line # 415 and the statement is replaced with the recommended way of rephrasing. 

‘Since gender-diverse persons make up such a small percentage of the population, it can be assumed that most of the academic literature on MHM is written from the perspective of cis women’. 

2. I think you would do well to read around the purpose of positionality statements and this would help to inform how to write one. I wasn’t looking for you just to state that you are Ethiopian but how that background and identity shapes and influences the research. For example, something like: I am a woman (?) who comes from Ethiopia, which is one of the countries studied in this paper. This gives me first-hand experience of what it’s like to be a menstruating schoolgirl in an LMIC. However, I am an outsider to the other countries studied in this paper, which may leave room for bias or misunderstandings in the interpretation of results. Something like this.

 Authors’ Response:

Dear reviewer the comment is well noted and the positionality statement in line #128-136, is rephrased as; 

“I (the first author) am a woman, a feminist, and an advocate for girls' education. I am currently pursuing a PhD in public health. I am from Ethiopia, which is one of the countries studied in this paper. This gives me first-hand experience of what it’s like to be a menstruating schoolgirl in an LMIC. However, I am an outsider to the other countries studied in this paper, which may leave room for bias or misunderstandings in the interpretation of results. In this systematic review, the researcher's standpoint influences the research approach and findings. This study advocates for accessible menstrual hygiene resources and aims to address the stigma surrounding menstruation. The conclusions are based on this perspective, and readers are encouraged to take this into account when interpreting the findings.”

3. Perhaps I’m being too cautious, but I still believe this could be interpreted as offensive to some readers ‘In certain religious texts, menstruation is often framed as making the menstruator impure or unclean, leading to menstrual restrictions, shame, and taboos’. To reiterate, so much religious text is down to interpretation, and this is what can cause the issues of negative views towards menstruation, not necessarily the words in the text itself. I don’t know if you’re religious or not, but this would be an instance where your positionality affects how you interpret data. For example, I’m not religious and therefore don’t understand or have association to any religious texts. I gave a lecture on MHM making the exact same point as you that religious texts e.g. The Quran says that menstruation is impure. This was of course from a translation from Arabic into English where meaning can get lost anyway. A Muslim student put their hand up to explain that this was a very negative interpretation of the text and that it doesn’t mean impure in a dirty way as was interpreted. I think it would be inoffensive but still make the same point to say ‘Certain religious texts can be interpreted to associate menstruation with impurity and uncleanliness, which leads to menstrual restrictions, shame, and taboos in some cultures’ you need a reference for this as well.

 Authors’ Response:

Dear reviewers I agree with comments. The statement in line Line #392-94: “In certain religious texts, menstruation is often framed as making the menstruator impure or unclean, leading to menstrual restrictions, shame, and taboos” 

Is replaced with 

‘Certain religious texts can be interpreted to associate menstruation with impurity and uncleanliness, which leads to menstrual restrictions, shame, and taboos in some cultures. And reference is cited.

---

## [Editor Report · Decision Letter 3]

8 Apr 2024

Menstrual hygiene management interventions and their effects on schoolgirls’ menstrual hygiene experiences in low and middle countries: A systematic review

PONE-D-23-35026R3

Dear Dr. Betsu,

We’re pleased to inform you that your manuscript has been judged scientifically suitable for publication and will be formally accepted for publication once it meets all outstanding technical requirements.

Kind regards,

Alison Parker

Academic Editor

PLOS ONE
---

## [Editor Report · Acceptance letter]

14 Jun 2024

PONE-D-23-35026R3 

PLOS ONE

Dear Dr. Betsu, 

I'm pleased to inform you that your manuscript has been deemed suitable for publication in PLOS ONE. Congratulations! Your manuscript is now being handed over to our production team.

Kind regards, 

on behalf of

Dr. Alison Parker 

Academic Editor

PLOS ONE